# Deoxypodophyllotoxin, a Lignan from *Anthriscus sylvestris*, Induces Apoptosis and Cell Cycle Arrest by Inhibiting the EGFR Signaling Pathways in Esophageal Squamous Cell Carcinoma Cells

**DOI:** 10.3390/ijms21186854

**Published:** 2020-09-18

**Authors:** Ah-Won Kwak, Mee-Hyun Lee, Goo Yoon, Seung-Sik Cho, Joon-Seok Choi, Jung-Il Chae, Jung-Hyun Shim

**Affiliations:** 1Department of Pharmacy, College of Pharmacy, Mokpo National University, Jeonnam 58554, Korea; rhkrdkdnjs12@mokpo.ac.kr (A.-W.K.); gyoon@mokpo.ac.kr (G.Y.); sscho@mokpo.ac.kr (S.-S.C.); 2College of Korean Medicine, Dongshin University, Naju, Jeonnam 58245, Korea; mhlee@dsu.ac.kr; 3College of Pharmacy, Daegu Catholic University, Hayang-Ro 13-13, Hayang-Eup, Gyeongsan-si, Gyeongbuk 38430, Korea; joonschoi@cu.ac.kr; 4Department of Dental Pharmacology, School of Dentistry and Institute of Oral Bioscience, BK21 Plus, Jeonbuk National University, Jeonju 54896, Korea; 5Department of Biomedicine, Health & Life Convergence Sciences, BK21 Four, College of Pharmacy, Mokpo National University, Jeonnam 58554, Korea

**Keywords:** deoxypodophyllotoxin, esophageal squamous cancer, apoptosis, EGFR

## Abstract

Deoxypodophyllotoxin (DPT) derived from *Anthriscus sylvestris* (L.) Hoffm has attracted considerable interest in recent years because of its anti-inflammatory, antitumor, and antiviral activity. However, the mechanisms underlying DPT mediated antitumor activity have yet to be fully elucidated in esophageal squamous cell carcinoma (ESCC). We show here that DPT inhibited the kinase activity of epidermal growth factor receptor (EGFR) directly, as well as phosphorylation of its downstream signaling kinases, AKT, GSK-3β, and ERK. We confirmed a direct interaction between DPT and EGFR by pull-down assay using DPT-beads. DPT treatment suppressed ESCC cell viability and colony formation in a time- and dose-dependent manner, as shown by MTT analysis and soft agar assay. DPT also down-regulated cyclin B1 and cdc2 expression to induce G2/M phase arrest of the cell cycle and upregulated p21 and p27 expression. DPT treatment of ESCC cells triggered the release of cytochrome c via loss of mitochondrial membrane potential, thereby inducing apoptosis by upregulation of related proteins. In addition, treatment of KYSE 30 and KYSE 450 cells with DPT increased endoplasmic reticulum stress, reactive oxygen species generation, and multi-caspase activation. Consequently, our results suggest that DPT has the potential to become a new anticancer therapeutic by inhibiting EGFR mediated AKT/ERK signaling pathway in ESCC.

## 1. Introduction

Esophageal cancer remains the sixth leading cause of cancer-related deaths, with a five-year survival rate of around 15–25% [1,2]. Esophageal cancer includes adenocarcinoma and squamous cell carcinoma, and of these, esophageal squamous cell carcinoma (ESCC) is the major type [3]. ESCC occurs at high incidence in China, Iran, East and Southern Africa [1,4,5]. The potential environmental risk factors of ESCC include smoking, alcohol, and hot drinks [1,2,4,6]. Treatments for ESCC include surgery, chemotherapy, or radiation therapy [1,2]. Although 5-fluorouracil and cisplatin have been a mainstay of cancer treatment, they have serious toxic side effects on both tumors and normal tissues [7,8,9]. A new non-toxic modality for preventing and treating ESCC, therefore, urgently needs to be developed.

Deoxypodophyllotoxin (DPT) is a precursor of podophyllotoxin isolated from the roots of *Anthriscus sylvestris* (L.) Hoffm [10] and has therapeutic pharmacological effects (Figure 1A). These include anti-inflammatory, antiplatelet aggregation, antiallergic, antiproliferation, antitumor, and antiviral properties [11]. A number of studies have reported that DPT induces apoptosis through cell cycle arrest and caspase-mediated pathways in prostate, gastric, and cervical cancer [11,12,13,14]. However, the question of whether DPT suppresses ESCC by induction of apoptosis has not been addressed.

It is well understood that signaling cascades from the epidermal growth factor receptor (EGFR/HER1/ERBB1) promote cell survival, proliferation, and invasiveness [15,16]. EGFR phosphorylation specifically triggers the activation of PI3K/AKT and mitogen-activated protein kinase (MAPK) kinase/ERK signaling pathway [15,16]. Active AKT and ERK both phosphorylate Bcl-2 associated agonist of cell death (Bad) and inhibit the apoptosis-inducing function of Bad [17]. Apoptosis is a form of programmed cell death that results in the removal of damaged cells [18,19]. There are two major apoptosis signaling pathways: the intrinsic (mitochondrial) and the extrinsic (death receptor-mediated) [18,19,20]. The intrinsic pathways are triggered by various cellular stresses [19]. This pathway regulates the Bcl-2 family to induce mitochondrial outer membrane permeation to spread cytochrome c (cyto c) [19,21]. Cyto c interacts with apoptotic protease-activating factor-1 (Apaf-1) and procaspase-9 to form an apoptosome [22]. Apoptosomes induce apoptosis by activating the downstream executioner caspases, including caspase-3, caspase-6, and caspase-7 [23]. Alternatively, the death ligand activates the death receptor to trigger the extrinsic pathway [24,25,26]. The death-inducing signaling complex activates caspase-8 and -10 [24,26]. Then activated caspase-8 cleaves Bid into truncated Bid (tBid) and induces downstream executioner caspase activation [18,20,27]. tBid also regulates the Bcl-2 family to promote mitochondrial outer membrane permeation [27].

In the present study, we examined the effect of DPT on ESCC cells. The results showed that DPT inhibited ESCC cell growth by targeting EGFR and related signaling, and induced cell cycle arrest and apoptosis through reactive oxygen species (ROS) production, mitochondrial membrane potential (MMP) depolarization, and multi-caspase activation in ESCC cells.

## 2. Results

### 2.1. DPT Targets EGFR and Regulates Cellular Signaling Pathways Associated with EGFR

To determine whether there is a direct interaction between DPT and EGFR, we conducted ex vivo pull-down assays using Sepharose 4B or DPT-Sepharose 4B beads in KYSE 30 and KYSE 450 cell lysates. As seen in Figure 1B, ex vivo pull-down assay and Western blotting demonstrated that DPT directly binds to EGFR, but DPT does not interact directly with AKT. To confirm the interaction of DPT with EGFR or AKT, we performed a kinase assay in combination with EGFR inhibitor gefitinib and AKT inhibitor MK-2206 as positive controls. Compared to the untreated control group, EGFR kinase activity was suppressed by DPT treatment (Figure 1C). However, when compared to the positive control (MK-2206), DPT treatment only slightly inhibited AKT1 and AKT2 kinase activity (Figure 1D, E). As shown in Figure 1F, DPT not only inhibited EGFR kinase activity, but also suppressed EGFR and its downstream activation of AKT/GSK-3β/ERK. Western blot assays indicated that the levels of phospho (p)-EGFR, p-AKT, p-GSK-3β, and p-ERK were suppressed following treatment with different concentrations of DPT, while their total amounts were not changed (Figure 1F). These results suggest that DPT binds to EGFR and thereby inhibits activation of EGFR and downstream proteins in ESCC cells.

### 2.2. DPT Inhibits the Growth of Human ESCC Cells

The cell viability of five types of ESCC cells and JB6 cells in the presence of DPT was evaluated by 3-(4,5-dimethylthiazol-2-yl)-2,5 diphenyltetrazolium bromide (MTT) assay. The cells were treated with DPT concentration 5, 7.5, and 10 nM for 24 h or 48 h. As shown in Figure 2A–E, the cell viabilities of ESCC cells were significantly decreased by DPT treatment in a time- and dose-dependent manner. IC_50_ values for DPT were approximately 7.64 nM (KYSE 30), 8.86 nM (KYSE 70), 8.92 nM (KYSE 410), 8.10 nM (KYSE 450), and 8.88 nM (KYSE 510) after 48 h of treatment, respectively. The results in Figure 2F indicate that 5–10 nM DPT didn’t affect the cell proliferation of JB6 (a mouse epidermal cell line) cells. In addition, we examined whether DPT would inhibit the colony formation capability of KYSE 30 and KYSE 450 cells. Figure 2G,H shows that DPT prominently reduced the number of KYSE 30 and KYSE 450 cell colonies.

### 2.3. DPT Causes G2/M Phase Accumulation and Apoptosis in KYSE 30 and KYSE 450 Cells

Cell cycle distributions were observed in KYSE 30 and KYSE 450 cells using Muse™ cell cycle reagent. Sub-G1 phase was <6% in untreated control groups, which increased to 35.67% and 34.77% after treatment of the cells with 10 nM concentration of KYSE 30 and KYSE 450 cells, respectively (Figure 3A,B); these responses were dose-dependent. As shown in Figure 3C, DPT caused dose-dependent accumulation of the percentage of the G2/M phase in ESCC cells. Proteins related to the G2/M phase of the cell cycle include p21, p27, cyclin B1, and cdc2, as shown in Figure 3D. p21 and p27 are cyclin-dependent kinase (CDK) inhibitors and regulate the cyclin-dependent kinase to stop cell cycle progression [28]. Cdc2-cyclin B (cell division control protein kinase 2), also known as, cdc2-cyclin B is a member of the CDK family connected with cell cycle control in eukaryotes [28]. p21 and p27 restrain the interaction of Cyclin B and cdc2, causing G2/M phase arrest [28]. Accordingly, the Western blot assay in Figure 3D confirmed increases of p21 and p27 protein expression and decreased expression of cyclin B and cdc2, as the DPT concentration increased. To estimate whether DPT can affect the apoptosis of ESCC cells, Annexin V/7-aminoactinomycin D (7-AAD) staining and flow cytometric assays were performed. The percentages of total apoptotic cells were 46.2 ± 5.8% (KYSE 30) and 44.9 ± 1.1% (KYSE 450) at 10 nM concentration, which were higher than seen in the control cells (Figure 4A,B).

### 2.4. DPT Induces Multi-Caspase Activity

To evaluate ESCC cell death by DPT treatment, a multi-caspase assay was performed, and the activity of multiple caspases (caspase-1, -3, -4, -5, -6, -7, -8, and -9) were measured by Muse™ Cell Analyzer (Figure 4C,D). Multi-caspase activity in the control group was 5.4% and 3.8% in KYSE 30 and KYSE 450, respectively. Total multi-caspase activity with 5, 7.5, and 10 nM DPT administration was 9.3%, 25.1%, and 49.9% in KYSE 30, and 8.8%, 11.8%, and 65.7% in KYSE 450 respectively. It was thus confirmed that the activation of multi-caspase increased concentration-dependently upon treatment with DPT in KYSE 30 and KYSE 450 cells. This suggests that DPT induces apoptosis through exogenous and endogenous pathways in KYSE 30 and KYSE 450 cells.

### 2.5. DPT Induces Apoptosis by Increasing Intracellular ROS Levels

In order to further investigate the mechanisms by which DPT promotes apoptosis of ESCC cells, we analyzed ROS levels after treatment with DPT (Figure 5A,B). Intracellular ROS production was detected following treatment with dimethyl sulfoxide (DMSO) or specified DPT concentrations (5, 7.5, and 10 nM) for 48 h. ROS levels increased significantly in a concentration-dependent manner in KYSE 30 and KYSE 450 cells (Figure 5A,B). To demonstrate whether this increase in ROS production played an important role in DPT-induced apoptosis, the ESCC cells were pretreated with N-acetyl-L-cysteine (NAC), a ROS inhibitor, for 3 h prior to treatment with DPT for 48 h. The results showed that DPT inhibited cell viability of KYSE 30 and KYSE 450 as expected, but NAC treatment (6 mM) attenuated the DPT-induced ESCC cell growth inhibition (Figure 5C). To further determine whether ROS involved in apoptosis induced by DPT, we treated cells with the ROS scavenger, NAC, alone or in a combination with DPT for 48 h in KYSE 30 and KYSE 450 cells. Annexin V/7-AAD data was consistent with multi-caspase activity, where apoptotic cell death was detected upon DPT treatment, and eliminated with the addition of NAC (Figure 5D,E). The apoptosis signal pathway was observed in DPT-treated cells alone, and it was subsequently abolished in cells pretreated with NAC. Endoplasmic reticulum (ER) stress typically induces apoptosis [29]. Expression of ER stress markers 78-kDa glucose-regulated protein (GRP78) and C/EBP homologous protein (CHOP; GADD153) are induced at high levels by ER stress [29,30]. By Western blotting, we found that the expression of ER stress markers, GRP78, and CHOP were increased in KYSE 30 and KYSE 450 cells treated with DPT (Figure 5F). Also, DPT upregulated the protein levels of death receptor (DR) 4 and DR5 in a dose-dependent manner in ESCC cells (Figure 5F). Collectively, the results demonstrated that DPT-induced apoptosis was related to ROS generation and ER stress marker increases.

### 2.6. DPT Influences MMP Dysfunction and Effects Mitochondria-Mediated Apoptosis Protein Expression in ESCC Cells

To find out whether DPT can induce apoptosis by disrupting MMP, KYSE 30, and KYSE 450 cells, they were stained with MitoPotential agent, a cationic and lipophilic dye, to monitor MMP. Percentages in the top left show depolarized/dead cells and percentages in the bottom left show depolarized/live cells. The results demonstrated that DPT treatment (10 nM) led to depolarization of the MMP of 50.8 ± 2.7% and 49.2 ± 7.7% in the KYSE 30 and KYSE 450 ESCC cells, respectively (Figure 6A,B). Next, we examined mitochondria-mediated apoptosis through the analysis of related proteins by Western blotting (Figure 6C). The expression of proteins belonging to the BCL-2 family, the antiapoptotic proteins Mcl-1, Bid, and Bcl-XL were decreased, and the apoptotic protein Bad exhibited increased expression, depending on the concentration of DPT. In addition, the expression of Apaf-1 and c-PARP were elevated, and the internal control used for all was actin. As apoptosis occurred in a dose-dependent manner following treating with DPT, cyto c from mitochondria was released into the cytosol. Concurrently, cyto c in mitochondria decreased. α-tubulin and COX4 were used as internal controls. Taken together, these results imply that DPT induced MMP dysfunction and mitochondria-mediated apoptosis.

## 3. Discussion

Investigation and development of new anticancer drugs is crucial for cancer treatment since the existing chemotherapeutic drugs, 5-fluorouracil and cisplatin, used for esophageal cancer, have severe side effects, including toxicity to normal cells, which greatly limits their use [31]. Therefore, it is essential to develop new drugs without toxicity and side effects for effective cancer treatment.

Botanical drugs and their derivatives have proven effective in the treatment of many types of cancer [31,32]. DPT, the precursor of podophyllotoxin, is a lignan that has various pharmacological effects [10,11]. In previous studies, DPT has been found to adjust cell growth, proliferation, cell cycle arrest, cancer cell invasion, and metastasis in several different types of cancer cells [12,13,14]. DPT is an aryltetralin cyclolignan structurally closely related to the lignan podophyllotoxin that contains four consecutive chiral centers and four almost planar fused rings and picropodophyllotoxin, an epimer of podophyllotoxin studied in the previous paper [33]. Since the structures of DPT and picropodophyllotoxin are similar, we experimented with the JNK/p38 MAPK pathway, but they were not the same signaling pathway. So, we proceeded with kinase screening for DPT. In the present study, we systematically evaluated the anti-ESCC effects of DPT in vitro and ex vivo. One of the significant findings from our study was that the mechanism of the apoptosis induced by DPT treatment included inhibition of EGFR kinase activity (Figure 1C). EGFR activates downstream molecules involved in the regulation of cell proliferation, survival, and differentiation [16]. The previous studies demonstrated that DPT induced apoptosis by inhibiting the IGF1R/PI3K/Akt signaling pathway in human non-small cell lung cancer and glioblastoma multiforme cells [34,35]. We wondered if DPT induces a similar pathway in ESCC cells as in previous studies. The results of this study showed that as the concentration of DPT increased, EGFR phosphorylation was inhibited, and consequently, the phosphorylation levels of downstream proteins (AKT, GSK-3β, and ERK) were reduced (Figure 1F). Thus, targeting EGFR, suppressing the AKT/GSK-3β/ERK signaling, might be a promising therapeutic strategy to overcome ESCC. Our study demonstrated mechanistic elements of DPT’s antitumor effect, but further studies are still needed. We found that DPT inhibited cell viability and elicited apoptosis in ESCC cells. We conducted the MTT assay to determine whether DPT could induce cytotoxicity in ESCC cells. When five different ESCC cell lines were treated with DPT, the cell viability rate was proportionally reduced with the increase of DPT concentration (Figure 2A–E). In JB6 cells, cell toxicity determined by MTT assay was not significant for any treatment group (Figure 2F). According to an earlier study, DPT (20 mg/kg) treatment caused a slight change in body weight of rodents in in vivo anticancer experiments, but did not show side effects such as diarrhea and skin rash [13,36]. We demonstrated that DPT inhibited the colony formation of KYSE 30 and KYSE 450 cells in dose-dependent manner (Figure 2G,H). The effects of DPT on the distribution of cell cycle phases in KYSE 30 and KYSE 450 were examined and these experiments demonstrated cell accumulation at the sub-G1 and G2/M phase in a dose-dependent manner (Figure 3A–C). Cyclin and CDK are well known to be major regulators of cell cycle progression [37]. CyclinB1 and cdc2 interact to form mitosis-promoting factor (MPF) that regulates the transition from G2 to M phase and plays an important role for mitotic entry during mitosis [38,39]. p21 and p27 suppress cyclin B/cdc2 to inhibit G2/M phase cell cycle progression [40]. We observed reductions in cyclin B1 and cdc2 expression levels (Figure 3D). On the other hand, the expression of p21 and p27 were dose-dependently increased by DPT treatment (Figure 3D). Apoptosis can be caused by intrinsic or extrinsic signals such as genotoxic stress or the binding of ligands to death receptors on the cell surface [18]. ROS can act as signaling molecules that induce oxidative stress, leading to increased cytotoxicity and apoptosis [41]. In our experiments, we found that DPT increases intracellular ROS levels in ESCC cells dose-dependently (Figure 5A,B). When NAC (ROS scavenger) was used prior to DPT treatment, it blocked ROS generation to prevent a decrease in cell viability (Figure 5C). The present results demonstrated that co-treat with NAC effectively inhibited DPT-induced apoptosis and multi-caspase activity in KYSE 30 and KYSE 450 cells (Figure 5D,E). Addition of antioxidant, NAC, with DPT suppressed the apoptosis in ESCC cell lines. DPT also induced generation of ROS and potent antiproliferative activity in KYSE 30 and KYSE 450 cell lines. These results suggest that DPT may have anticancer effect on ESCC cells through ROS-mediated apoptosis. In other words, it is noteworthy that ROS plays an important role in DPT-induced apoptotic mechanisms. Apoptosis is composed of two pathways; intrinsic and extrinsic. The intrinsic pathway is regulated by the Bcl-2 family protein [18]. The antiapoptotic protein group includes Bcl-2, Bcl-XL, and Mcl-1 and the pro-apoptotic protein group includes truncated Bid, Bim, Bax, and Bad [18]. DPT caused apoptosis by increasing the loss of MMP (Figure 6A,B) and through up- or down-regulation of proteins closely related to the intrinsic (mitochondrial) pathway (Figure 6C). Apoptosis by DPT treatment results in cyto c release from mitochondria to the cytoplasm, and induces Apaf1 and cleavage of PARP (Figure 6C). Caspases play an important role in the apoptosis mechanism as initiators (caspase-2, -8, -9 and -10) and executors (caspase-3, -6 and -7) [18]. Our results show that treatment with DPT activated multiple caspases and induced apoptosis in ESCC cells (Figure 4C,D).

Based on our findings, DPT induces apoptosis by inhibiting the phosphorylation of EGFR directly in ESCC cells, and thus illustrates the mechanism behind its antitumor effects. The antitumor efficacy of DPT occurs through the inhibition of EGFR activity, and connotes that it was a primary target of DPT. We therefore suggest that DPT might represent a new generation of antitumor drugs for treating ESCC.

## 4. Materials and Methods

### 4.1. Reagents and Antibodies

DPT was prepared by Professor Goo Yoon according to previous reports [42]. RPMI1640 medium, phosphate-buffered saline, fetal bovine serum, penicillin/streptomycin, and trypsin were purchased from Hyclone (Logan, UT, USA). MEM medium was purchased from Corning (Corning, NY, USA). DMSO, MTT, and Basal Medium Eagle were purchased from Sigma Chemical Company (St. Louis, MO, USA). Gefitinib was purchased from Cayman Chemical (Ann Abor, MI, USA). Primary antibodies against cyclin B1, cdc2, p21, p27, β-actin, GRP78, CHOP, DR4, DR5, Bcl-XL, Mcl-1, Bid, Bad, cyto c, α-tubulin, COX4, apoptotic protease activating factor-1 (Apaf-1), and c-PARP were obtained from Santa Cruz Biotechnology (Santa Cruz, CA, USA). Antibodies against p-EGFR (Tyr1068), EGFR, p-GSK-3β (Ser9), GSK-3β, p-ERK (Thr202/Tyr204), ERK, p-AKT (Ser473), and AKT were purchased from Cell Signaling Biotechnology (Beverly, MA, USA).

### 4.2. Cell Culture

Human esophageal squamous cell carcinoma cell lines (KYSE 30, KYSE 70, KYSE 410, KYSE 450, and KYSE 510) were acquired from the Type Culture Collection of the Chinese Academy of Sciences (Shanghai, China) and the American Type Culture Collection (ATCC, Rockville, MD, USA) and maintained in RPMI1640 medium (Hyclone, Logan, UT, USA), supplemented with 10% fetal bovine serum and 1% penicillin/streptomycin. JB6 (a mouse epidermal cell line) cells were purchased from American Type Culture Collection (ATCC; Rockville, MD, USA). JB6 cells were cultured in MEM medium (Corning, NY, USA) containing 10% FBS and antibiotics. The cells were incubated in a humidified atmosphere incubator containing 5% CO_2_ at 37 °C. The cell culture cycle was 3 days. Experiments were conducted at 24 h and 48 h after drug treatment. We stored cells from p1 to p3. We ran the experiment from p4 to p15 and changed cells. Before proceeding with the experiment, we conducted mycoplasma testing using a mycoplasma PCR detection kit.

### 4.3. Cell Viability Assay

ESCC cells (KYSE 30, KYSE 70, KYSE 410, KYSE 450, and KYSE 510) and JB6 cells were seeded in 96-well plates at 2.75 × 10^3^, 10 × 10^3^, 2.5 × 10^3^, 3.5 × 10^3^, 5.5 × 10^3^, and 8 × 10^3^ per 100 μL and incubated for 24 or 48 h after treatment with DPT (5, 7.5, and 10 nM). At the end of each treatment time, 30 μL MTT was added to each well. The plates were incubated for 1 h to 1 h 30 min at 37 °C and then shaken. The absorbance of each well at 570 nm was measured using a spectrophotometer (Thermo Fisher Scientific, Vantaa, Finland).4.4. Cell Cycle Assay

### 4.4. Cell Cycle Assay

Cells (KYSE 30 and KYSE 450) at a density of 7.5 × 10^4^/well and 10.5 × 10^4^/well were cultivated in 6-well plates for 48 h with the indicated concentrations of DPT (5, 7.5, and 10 nM). The cells were trypsinzed and harvested. Both cells were washed three times and fixed with 70% ethanol overnight. After washing in cold phosphate-buffered saline (PBS), the cells in 100 μL PBS were then stained with 150 μL Muse™ Cell Cycle Reagent for 30 min. The stained cells were measured by Muse™ Cell Analyzer (Merck Millipore, Billerica, MA, USA).

### 4.5. Cell Apoptosis Analysis

KYSE 30 and KYSE 450 cells were exposed to indicated concentrations of DPT for 48 h. The cells were collected by RPMI-1640 medium. 60 μL cells were added with 100 μL Muse™ Annexin V & Dead Reagent and then incubated for 20 min at room temperature. The samples were analyzed using Muse™ Cell Analyzer (Merck Millipore, Billerica, MA, USA), and 10,000 events from each sample were obtained to ensure suitable data.

### 4.6. ROS Assay

Cells (KYSE 30 and KYSE 450) were seeded into 6-well plates at a density of 7.5 × 10^4^/well and 10.5 × 10^4^/well. Afterward, the cells were treated with DPT concentration. The cells were stained with 190 μL Muse™ Oxidative Stress Reagent working solution and incubated 37 °C for 30 min. The levels of ROS in ESCC cells were measured by using Muse™ Cell Analyzer (Merck Millipore). To confirm the generation of intracellular ROS, cells were pretreated for 3 h with 6 mM N-acetyl-L-cysteine (NAC; a hydrogen peroxide scavenger) as a negative control prior to treatment with DPT.

### 4.7. Measurement of Mitochondrial Membrane Potential (MMP) Assay

MMP was determined using Muse™ MitoPotential Kit (Merck Millipore). Briefly, after treating the cells with 5, 7.5, and 10 nM of DPT for 48 h, cells were stained with Muse™ MitoPotential working solution for 20 min at 37 °C. After staining with 7-AAD for 5 min, the MMP was analyzed by Muse™ Cell Analyzer (Merck Millipore).

### 4.8. Multi-Caspase Assay

To determine apoptosis by activation of multiple caspases (caspase-1, -3, -4, -5, -6, -7, -8 and -9), the Muse™ Multi-caspase assay kit (Merck Millipore) was used. Cells were treated with the specified concentration of DPT. After harvesting the cells, cells were washed with 1X caspase buffer. 5 μL Muse™ Multi-Caspase reagent working solution was added to the cells and incubated for 30 min at 37 °C. 125 μL of Muse™ Caspase 7-AAD working solution was added in each tube and kept incubated for 5 min at room temperature. The data were detected using the Muse™ Cell analyzer (Merck Millipore).

### 4.9. Cytosol, Mitochondria Fraction

The cells were treated with 0, 5, 7.5, and 10 nM DPT for 48 h, then cells were harvested and suspended with the plasma membrane extraction buffer. Samples were added with 0.1% of digitonin and centrifuged for 5 min at 13,000 rpm. The lysates (cytoplasmic protein extract) were used for Western blotting. The pellets were washed in plasma membrane extraction buffer and re-centrifuged. The supernatant was removed and the pellet was resuspended with plasma membrane extraction buffer containing 0.5% of Triton X-100 and incubated for 10 min in ice. The samples were centrifuged at 13,000 rpm for 30 min at 4 °C and the lysates (membrane protein fraction) were collected for Western blotting.

### 4.10. Western Blotting

Protein concentrations were determined by DC Protein Assay (Bio-RAD, Hercules, CA, USA). The protein extracts were separated on 8~15% SDS-PAGE, and transferred onto polyvinylidene fluoride membranes. The membranes were saturated with 3% skim milk, incubated overnight at 4 °C with primary antibodies (1:1000), washed, and then incubated for 2 h with secondary antibodies, i.e., peroxidase-conjugated goat anti-mouse, anti-goat or anti-rabbit immunoglobulin G (Santa Cruz Biotechnology, 1:5000 to 1:10,000). Signals were revealed by autoradiography with the enhanced chemiluminescence detection kit (iNtRON Biotechnology, Korea) using ImageQuant LAS 500 (GE Healthcare, Uppsala, Sweden). Western bands were determined by densitometry of bands using Image J software.

### 4.11. Kinase Assay

EGFR kinase activity was determined with ADP-Glo™ kinase assay (Promega, Madison, WI, USA). The principle of this kinase assay is to assess kinase activity by quantifying the rest of ATP after a kinase reaction, which is decided by a luciferase-catalyzed response. The EGFR reaction and the ensuing luciferase response were performed in a 384-well plate. Solvent with EGFR (1.8 ng/μL, final concentration), substrates (0.2 μg/μL), DPT, gefitinib (1 μM), kinase buffer [40 mM Tris (pH 7.5), 20 mM MgCl_2_, 0.1 mg/mL BSA, 50 μM dithiothreitol, 2 mM MnCl_2_, and 100 μM sodium vanadate] were added to each well. Reactions in each well were started immediately by adding ATP and kept going for an hour under room temperature (22–25 °C). Kinase reactions were performed with a volume of 5 μL. 5 μL of ADP-Glo reagent (ADP-Glo kinase assay kit; Promega) was added into each well to stop the reaction and remove the rest of ADP within 40 min. Finally, 10 μL of kinase detection reagent was added into the well and incubated for 1 h to generate a luminescence signal. Luminescence was detected with a Centro LB 960 microplate luminometer (Berthold Technologies, Bad Wildbad, Germany) for 0.5 s.

### 4.12. Pull Down Assay

KYSE 30 and KYSE 450 cell lysates (1000 μg) were reacted with DPT-conjugated Sepharose 4B or Sepharose 4B beads in reaction buffer [50 mM Tris (pH 7.5), 5 mM EDTA, 150 mM NaCl, 1 mM/L dithiothreitol, 0.01% Nonidet P-40, 2 μg/mL bovine serum albumin, 0.02 mM phenylmethylsulfonyl fluoride, and 1X proteinase inhibitor] for overnight incubation with smooth rocking at 4 °C. The beads were washed six times with washing buffer [50 mM Tris (pH 7.5), 5 mM EDTA, 150 mM NaCl, 1 mM dithiothreitol, 0.01% Nonidet P-40, and 0.02 mM phenylmethylsulfonyl fluoride] and proteins bound to the beads were observed using Western blot analysis.

### 4.13. Statistical Analysis

All statistical analyses were performed using Prism 5.0 statistical package. Data were expressed as mean ± standard deviation (SD) otherwise specified. Multiple data statistical analysis was determined by one-way ANOVA. Comparison of two means was done by Student’s *t*-test. Mean values of * *p* < 0.05, ** *p* < 0.01, and *** *p* < 0.001 were considered statistically significant.

## Figures and Tables

**Figure 1 ijms-21-06854-f001:**
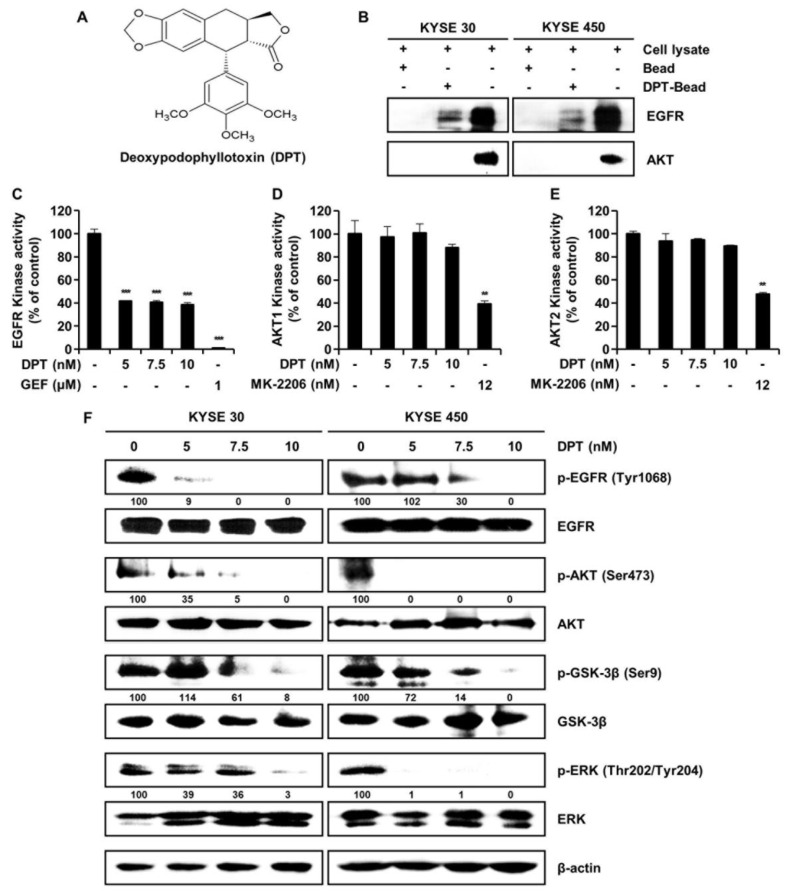
EGFR interacts with deoxypodophyllotoxin (DPT). (**A**) Chemical structure of DPT. (**B**) KYSE 30 and KYSE 450 cell lysates were mixed with DPT-conjugated Sepharose 4B beads or with Sepharose 4B beads alone, and the pulled-down proteins were measured using Western blotting. (**C**) EGFR kinase activity of DPT by ADP-Glo kinase assay. Gefitinib, a selective EGFR kinase inhibitor, was used as positive controls. Data is shown as mean ± standard deviation (SD) and performed as three independent experiments. *** *p* < 0.001 compared with control. (**D**,**E**) The effects of DPT on AKT1 and AKT2 kinase activity by ADP-Glo kinase assay. MK-2206, a selective AKT1/2 inhibitor, was used as a positive control. Data are mean ± SD of 3 experiments performed in triplicate. ** *p* < 0.01. (**F**) KYSE 30 and KYSE 450 cells were treated with various concentrations of DPT (0, 5, 7.5, and 10 nM) for 48 h. The phosphorylation of EGFR, AKT, GSK-3β, and ERK was identified by Western blotting. β-actin was used as a loading control.

**Figure 2 ijms-21-06854-f002:**
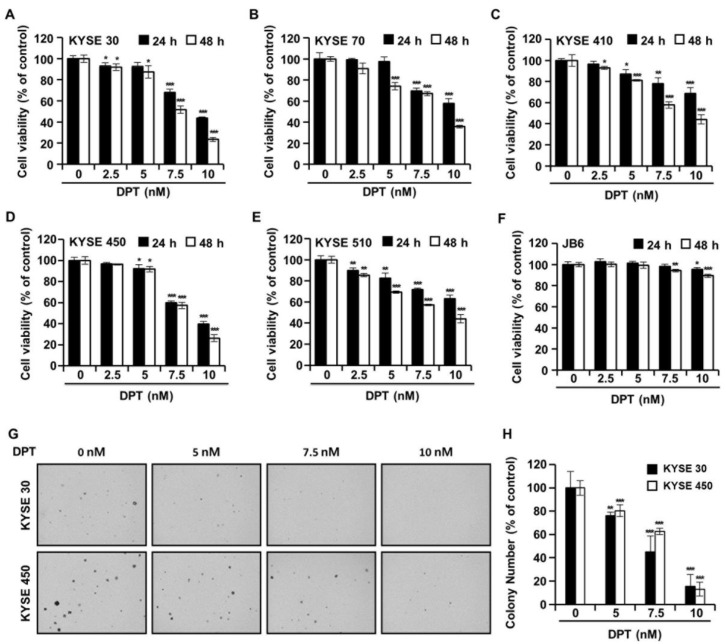
The effects of DPT on cell viability and anchorage-independent growth capability in ESCC cells. (**A**–**F**) Cell viability was determined after treatment with 0, 5, 7.5, and 10 nM of DPT for 24 and 48 h in JB6 cells and various ESCC cell lines, including KYSE 30, KYSE 70, KYSE 410, KYSE 450, and KYSE 510. MTT assay was used to assess JB6 and ESCC cell viability. (**G**,**H**) Cell anchorage-independent growth capability was evaluated by soft agar assay. KYSE 30 and KYSE 450 cells were incubated with DMSO or DPT (5, 7.5, and 10 nM) for 2 weeks. All data are shown as mean ± SD from three independent experiments. * *p* < 0.05, ** *p* < 0.01, and *** *p* < 0.001 compared to cells without DPT treatment.

**Figure 3 ijms-21-06854-f003:**
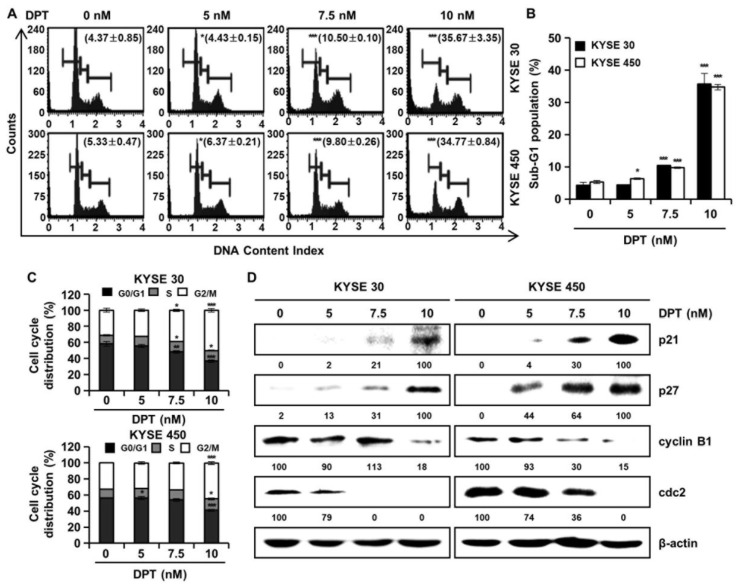
DPT induces G2/M phase cell cycle arrest in ESCC cells. KYSE 30 and KYSE 450 cells were treated with the indicated concentrations (0, 5, 7.5, and 10 nM) for 48 h. The cells were stained by Muse™ cell cycle reagent. (**A**) Representative histograms of the cell cycle. (**B**) Percentage of sub-G1 population. (**C**) Percentage of G0/G1, S, and G2/M phases. (**D**) The protein lysates were subjected to Western blot analysis with antibodies against p21, p27, cyclin B1, and cdc2 protein. β-actin was used as the loading control. All values are graphed as the means ± SD (* *p* < 0.05, ** *p* < 0.01, and *** *p* < 0.001 as compared to untreated controls).

**Figure 4 ijms-21-06854-f004:**
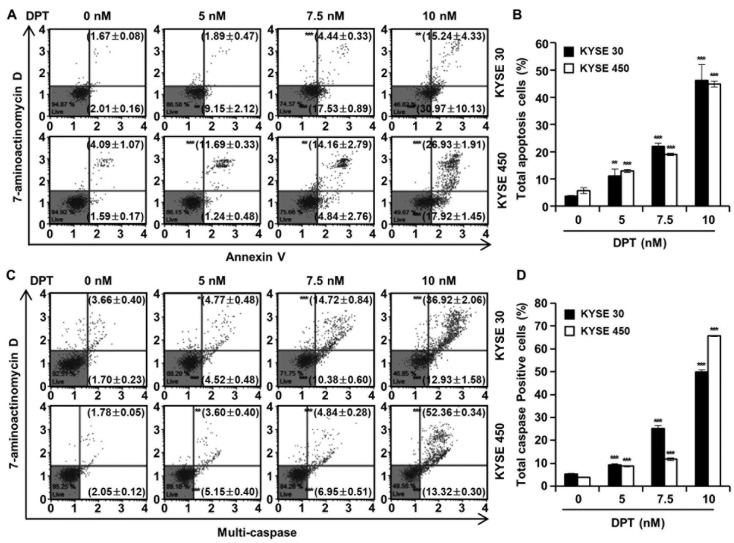
Effect of DPT on apoptosis and multi-caspase activity in ESCC cells. KYSE 30 and KYSE 450 cells were treated with DPT at 0, 5, 7.5, and 10 nM for 48 h. (**A**,**B**) Range of apoptosis decided by quantification of annexin V/7-aminoactinomycin D (7-AAD) stained cells. (**C**,**D**) Multi-caspases activity were measured by Muse™ Multi-caspase Kit. The plots illustrate the effect of DPT treatment on the ESCC cell lines indicated. Representative annexin V and multi-caspase dot plots are presented of three replicates. * *p* < 0.05, ** *p* < 0.01, and *** *p* < 0.001, significantly different from control group.

**Figure 5 ijms-21-06854-f005:**
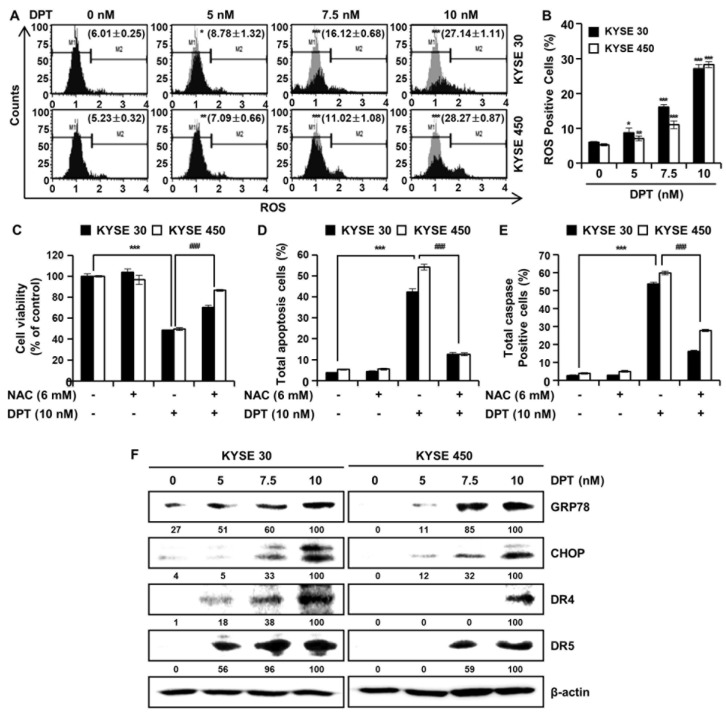
DPT induces ROS production in ESCC cells. Cells were treated with 0, 5, 7.5, and 10 nM of DPT for 48 h and were then harvested. (**A**,**B**) Following incubation with Muse™ Oxidative Stress Reagent working solution for 30 min, ROS generation was analyzed by Muse™ Cell Analyzer. ROS scavenger, N-Acetyl-L-Cysteine (NAC), inhibits DPT-induced apoptotic cell death. ESCC cell lines were treated with DMSO, 6 mM NAC, 10 nM of DPT, NAC with DPT for 48 h. (**C**) Cell viability was measured by MTT assay. (**D**) Apoptotic cell death was analyzed annexin V/7-AAD using Muse™ Cell analyzer. (**E**) Multi-caspase activity was detected by Muse™ Multi-caspase assay kit by Muse™ Cell analyzer. Data are the mean ± SD (*n* = 3 in each group). * *p* < 0.05, ** *p* < 0.01, and *** *p* < 0.001 vs. the control group, **###**
*p* < 0.001 vs. DPT treated group. (**F**) Equivalent protein amounts from KYSE 30 and KYSE 450 cell cultures were lysed, size-fractionated, and subjected to Western blot analysis with antibodies against GRP78, CHOP, DR4, and DR5. β-actin was used as the loading control.

**Figure 6 ijms-21-06854-f006:**
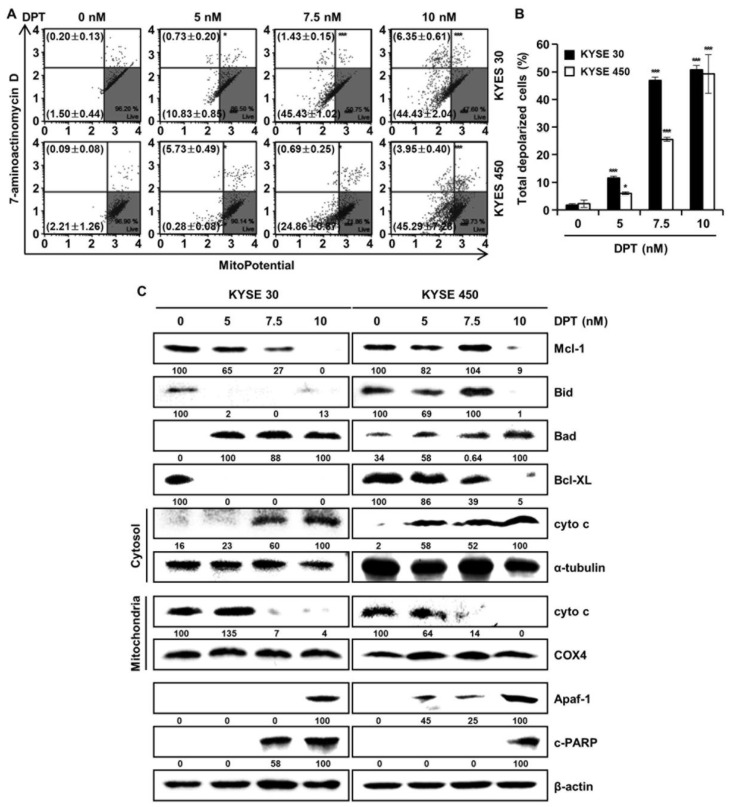
DPT-induced apoptosis in ESCC cells is mitochondria-mediated. Cells were treated with different concentrations of DPT for 48 h. (**A**,**B**) MMP (Δψ) analysis of KYSE 30 and KYSE 450 cells to examine the dose-dependent effect of DPT on mitochondrial dysfunction. Cells were stained with Muse™ Mitopotential working solution, and fluorescence intensity was assessed using Muse™ Cell Analyzer. * *p* < 0.05, and *** *p* < 0.001, significantly different compared with the control group. (**C**) Mcl-1, Bid, Bad, Bcl-XL, cyto c, α-tubulin, COX4, Apaf-1, and c-PARP expression in KYSE 30 and KYSE 450 cells were analyzed by Western blot. β-actin served as the loading control.

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
