# Peer review of "Deoxypodophyllotoxin, a Lignan from Anthriscus sylvestris, Induces Apoptosis and Cell Cycle Arrest by Inhibiting the EGFR Signaling Pathways in Esophageal Squamous Cell Carcinoma Cells"

_ijms, 2020, doi:10.3390/ijms21186854_

Round 1
Reviewer 1 Report
Ref: ijms-928137
Title: Deoxypodophyllotoxin, a lignan from Anthriscus sylvestris, induces apoptosis and cell cycle arrest by inhibiting the EGFR signaling pathways in esophageal squamous cell carcinoma cells. (Journal: IJMS)
Recommendation: Major Revision
Comments:
- The paper is very similar to previous work of the Kwak’s team (similar methodology etc.) - Kwak AW, Yoon G, Lee MH, Cho SS, Shim JH, Chae JI. Picropodophyllotoxin, an Epimer of Podophyllotoxin, Causes Apoptosis of Human Esophageal Squamous Cell Carcinoma Cells Through ROS-Mediated JNK/P38 MAPK Pathways. Int J Mol Sci. 2020;21(13):4640. Published 2020 Jun 30. doi:10.3390/ijms21134640 The Authors should provide the information how these chemicals differ from each other, make a comparison etc.
- The Authors indicate that a new drug for esophageal cancer should have no side effects such as healthy cells toxicity. The Authors should provide an information how DPT acts on primary cells e.g. human primary esophageal epithelial cells.
- The western blots should be quantified and showed also as a % of the control.
- Some WB have bad quality, e.g. KYSE 30 – AKT, KYSE 450 – EGFR (Fig. 1); KYSE 450 – p21 (Fig. 3).
- Fig.1 description – there should be an information about positive controls.
- There should be precise information about conducting cell culture (in vitro days, passages, mycoplasma testing).
- Please provide antibodies dilution in western blot description.
- Please follow the latest nomenclature rules, e.g.:
https://www.ncbi.nlm.nih.gov/genome/doc/internatprot_nomenguide/
https://academic.oup.com/molehr/pages/Gene_And_Protein_Nomenclature

Reviewer 2 Report
The authors in their research article describe the anti-proliferative and pro-apoptotic activity of deoxypodophyllotoxin (DPT) toward esophageal squamous cell carcinoma cells. DPT inhibited cell proliferation, induced G2/M cell cycle arrest and apoptosis through the mitochondrial pathway. Furthermore, DPT inhibited EGFR mediated AKT and ERK signaling, increased ROS generation and endoplasmic reticulum stress.
Overall, the study is a concise analysis of the anti-proliferative activity of DPT. The research presents interesting new data, adding to the present knowledge regarding the activity of DPT. My comments for the authors are listed below:
It would be beneficial to add histograms for the cell cycle analysis. How did the authors analyze the data from this experiment. Usually the sub-G1 population is included in the the total cell count, which is 100%. In the authors’ graphs the sub-G1 population, at the highest concentration, is around 35% which would give, along with G0/G1, S and G2/M, a total cell population count of 135%.
The authors conclude that DPT induces apoptosis through ROS generation. To confirm this, the influence of NAC pre-treatment should be evaluated on apoptosis induction, not only on cell viability.
I suggest restructuring the results section. The first paragraph should include results regarding activity analysis of the compound and this should be then followed by experiments related to the mechanism of action of the compound. I would also suggest transferring the results regarding caspase activation to the section regarding apoptosis induction and grouping them with Annexin V analysis in a separate Figure.
There is little discussion regarding previous studies related to the activity of DPT and results associated with the presented research, e.g., effects of DPT on PI3K/Akt signaling inhibition (Wang et al., 2019) or IGF1R/Akt signaling (Park et al., 2018).
What was the source of DPT used in the study? The authors should add this information in the Materials and Methods section.
The abstract requires corrections. In line 33, the authors mention that ‘DPT down-regulated p21, p27’, the results show the opposite. In line 36, the authors state that DPT induces ‘apoptosis by up regulation of related proteins’, some of the related proteins were also down-regulated.
Round 2
Reviewer 1 Report
The Authors answered all my questions. In my opinion the manuscript is ready for publication.
Author Response
˜ Reviewer #1 Question:
The Authors answered all my questions. In my opinion the manuscript is ready for publication.
â–¶ Response to Question:
Thank you.

Reviewer 2 Report
In the revised version of the manuscript the authors have acknowledged most of my questions. I have some additional comments for the authors regarding their revision, which are listed below:
- Figure 3. The added histograms to the cell cycle distribution analysis add clarity to the presented graphs. However, since the histograms do not show the percentage of the cell populations in the particular phases of the cell cycle, I would also include the graphs showing this information (previous Figure 3 A), which the authors did not include in the revised version.
- The authors did not address the question regarding discussing previous studies related to the activity of DPT and results associated with the presented research. This should be addressed or explained if the authors do not find the addition of this information relevant.
- The inserted fragments in the revised manuscript require language revision.
